# Mapping the Glymphatic Pathway Using Imaging Advances

Rajvi N. Thakkar [1], Ivelina P. Kioutchoukova [1], Ian Griffin [1], Devon T. Foster [2], Pratiksha Sharma [3], Eduardo Molina Valero [1] and Brandon Lucke-Wold [4,*]

1   College of Medicine, University of Florida, Gainesville, FL 32610, USA; r.thakkar@ufl.edu (R.N.T.)
2   College of Medicine, Florida International University, Miami, FL 33199, USA
3   Thermo Fisher Scientific Inc., Boston, MA 02210, USA
4   Department of Neurosurgery, University of Florida, 1600 SW Archer Rd., Gainesville, FL 32610, USA
*   Correspondence: brandon.lucke-wold@neurosurgery.ufl.edu

**Abstract:** The glymphatic system is a newly discovered waste-clearing system that is analogous to the lymphatic system in our central nervous system. Furthermore, disruption in the glymphatic system has also been associated with many neurodegenerative disorders (e.g., Alzheimer's disease), traumatic brain injury, and subarachnoid hemorrhage. Thus, understanding the function and structure of this system can play a key role in researching the progression and prognoses of these diseases. In this review article, we discuss the current ways to map the glymphatic system and address the advances being made in preclinical mapping. As mentioned, the concept of the glymphatic system is relatively new, and thus, more research needs to be conducted in order to therapeutically intervene via this system.

**Keywords:** glymphatic system; magnetic resonance imaging (MRI) sequencing; imaging techniques; neurological diseases

## 1. Introduction

The buildup of metabolic waste in the brain negatively affects neural functions, possibly even leading to the development of neurodegenerative diseases if not removed. The glymphatic system, analogous to the lymphatic system of the central nervous system (CNS), directs interstitial fluid movement and waste clearance in the brain to prevent such consequences [1]. These perivascular spaces are surrounded by astrocytic end-feet, which express aquaporin-4 water channels (AQP4) to facilitate the transportation of nutrients. With the help of AQP4 water channels, the exchange of cerebrospinal fluid (CSF) into the interstitial fluid (ISF) occurs to assist in lymphatic drainage and waste removal. This exchange allows the proper elimination and reabsorption of solutes and metabolites, fluid and electrolyte balance, and internal body pressure regulation. The drainage is driven in the direction of the CSF into the sinuses of the brain's dura mater. Lymphatic vessels, including meningeal lymphatic vessels, in the dura mater absorb CSF and ISF through the glymphatic system and transport the fluids to the base of the skull for extracranial drainage. The awake brain produces an increased buildup of proteins, weakening the glymphatic system's activity during wakefulness. The increased activity during sleep allows the necessary removal of harmful substances [2].

This newly discovered perivascular waste clearance system, despite its impressive results, has yet to see wide clinical application. This paper will explore various imaging techniques used to map the glymphatic system. Among the methods for mapping this pathway is the use of T1-weighted contrast-enhanced Magnetic resonance imaging (MRI). In T1-weighted MRI, gadolinium-based contrast agents can help map the glymphatic system by tracking CSF via intrathecal and intravenous administration [3]. This method has primarily been used to study chronic neurodegenerative diseases. However, disruption of the glymphatic system has also been observed in acute neurological complaints such

as traumatic brain injury (TBI) and subarachnoid hemorrhage. After a TBI, increased aggregates of the tau protein were seen, with evidence of a defective glymphatic system [4]. By using T1-weighted MRI, we can map the consequences of TBI, among other injuries, thereby having a clearer understanding of the onset of neurodegenerative diseases.

Glymphatic flow is facilitated through the perivascular system and is an important feature of the glymphatic system. The use of diffusion MRI techniques can help map this flow and thereby broaden the understanding of the glymphatic system's mechanism. This flow can then be further studied with diffusion tensor imaging (DTI), which can 3D map the flow pathway [5]. DTI can even evaluate patients with TBI, allowing for the quantification of the glymphatic pathway in both chronic- and acute-illness patients [6]. DTI shows the movement of water molecules and can be used to quantify the diffusivity of individuals, thus creating a better understanding of a patient's glymphatic system [3].

Single-photon emission computed tomography/computed tomography (SPECT/CT) is another imaging technique that provides an understanding of the glymphatic drug delivery pathway. The kinetics of CSF tracer flow is limited in other imaging techniques due to their limited spatial detail. SPECT/CT imaging allows the quantification of a tracer's clearance from the site of injection and enables the use of smaller doses of the tracer compared with contrast-enhanced MRI. This can then provide more knowledge about the impact of neurodegenerative diseases on glymphatic clearance [7].

Ultrafast magnetic resonance encephalography (MREG) can help image the entire brain in 100 ms and show the pulsation of CSF. By doing so, the glymphatic CSF flow pulsations can be better mapped and studied through MREG wave analysis. Mapping CSF flow will also help understand the decline of the glymphatic system and the onset of neurodegenerative diseases during the onset of aging [8].

Fluorescence microscopy and macroscopic imaging techniques are further imaging techniques that can help map the glymphatic system. The glymphatic system is highly active during the sleep state or under anesthesia when most waste clearance occurs while being suppressed during the awake state. Light sheet fluorescence microscopy can be used to detect these "sleep" and "awake" states of the glymphatic system. This imaging technique is particularly advantageous due to its ability to produce high-resolution imaging of the brain in a single session [9]. Additionally, transcranial glymphatic flow can be mapped with fluorescent macroscopy by delivering fluorescent tracers and using LED illumination for fluorophore excitation. In doing so, CSF flow in the glymphatic system can be further mapped [10].

These various imaging techniques discussed in this paper all serve to create a better understanding of the newly discovered glymphatic system (Figure 1). Considering the system's role in waste clearance and limiting the onset of neurodegenerative diseases, it is important to have a better understanding of its mechanism. Through these numerous techniques, different aspects of the glymphatic system can be mapped, including CSF flow, brain pulsations, drug delivery clearance, and the quantification of protein accumulation. All the techniques discussed in this paper serve to enhance the complicated pathway of the glymphatic system.

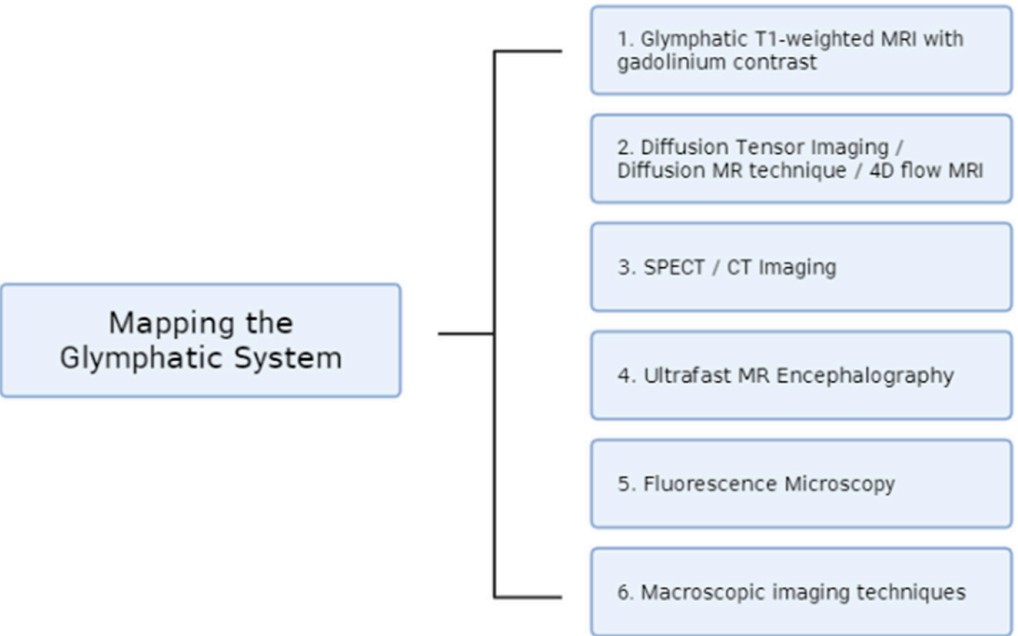

**Figure 1.** Summary of the imaging advances used for mapping the glymphatic system that are covered in this review.

## 2. Imaging Techniques

### 2.1. Glymphatic T1-Weighted MRI with Gadolinium Contrast

Gadolinium is a paramagnetic contrast agent that improves the visualization of CSF distribution in MRI. It is the most common contrast agent used in T1-weighted MRI to map the glymphatic pathway. Gadolinium-based contrast agents (GBCAs) can be administered both intravenously and intrathecally, with intravenous methods being well tolerated at a greater range of doses [11–13].

a.    Intrathecal gadolinium enhancement of glymphatic T1 MRI:

Building off of their seminal work to discover the glymphatic system, Iliff et al. adapted MR cisternography techniques to successfully analyze the kinetics of CSF flow and layout of the glymphatic system in the rat brain. The team reported that intrathecal contrast agents move via bulk flow through several influx nodes along the surface and penetrate para-arterial pathways into the brain parenchyma. Glymphatic flux was observed to occur through bulk flow, independent of molecular weight, in the proximal perivascular channels and subarachnoid spaces. Conversely, the interstitial uptake of a small-molecular-weight GBCA, gadolinium-diethylenetriaminepentaacetate (Gd-DTPA), was more rapid compared with a large-molecular-weight polymeric gadolinium-chelate in the glymphatic system [14]. The human glymphatic pathway has also been mapped using GBCA in conjunction with T1 MRI. One study demonstrated brain-wide distribution and enhancement in serial T1-weighted MRI when gadobutrol was injected into the subarachnoid compartments of human subjects, noting a similar flow of the contrast agents to what was previously observed in the rodent brain [15]. Studies on the glymphatic system using MRI with intrathecal GBCAs have confirmed that the system is most active during sleep and that it is impaired in neurodegenerative diseases including Alzheimer's disease and Parkinson's disease [16,17]. It follows that MRI using GBCAs may one day be incorporated into screening criteria for sleep disorders and dementia.

Clinically, T1 MRI enhancement with small doses of intrathecal Gd-DTPA has been well tolerated for diagnostic use in idiopathic normal-pressure hydrocephalus and aqueduct stenosis and for application to locate sites of CSF leakage to be surgically repaired in patients with idiopathic intracranial hypotension and CSF rhinorrhea [16,18,19]. However, intrathecal gadolinium is generally contraindicated outside of these specific cases, as the

accidental administration of large doses in humans has precipitated encephalopathy with loss of consciousness, seizure, and death [20]. Furthermore, intrathecal GBCAs can lead to the accumulation of gadolinium deposits in the brain, causing cognitive impairment [21]. MRI with intrathecal GBCAs is also being used to study the glymphatic system in animal models of stroke and neurotrauma.

One study injected gadolinium chelate into the cisterna magna of male Swiss mice to investigate the impact of stroke subtypes on in vivo glymphatic perfusion, reporting that subarachnoid hemorrhage (SAH) and the acute phase of ischemic stroke both severely impaired the glymphatic system [22]. This was not the case for carotid ligature or intracerebral hemorrhage, which was also induced by this team. Interestingly, the same study found that this effect can be reversed in SAH with intracerebroventricular injection of tissue-type plasminogen activators to clear the perivascular space. Recanalization of the arteries with thrombolytics also restored the function of the glymphatic system after embolic ischemic stroke. This suggests that GBCAs may be useful to confirm both arterial patency and perivascular clearance following an acute intervention.

As our understanding of CSF flow along the glymphatic pathway improves, we will likely see more targeted therapies being administered intrathecally. GBCAs may therefore see additional utility as paramagnetic tracers that are conjugated to these emerging therapies to analyze their distribution to intended regions of the brain. MRI with an intra-cisternal Gd-DTPA was also found to be sensitive to glymphatic disruptions via mild traumatic brain injury (mTBI) in rats [23]. Thus, intrathecal GBCAs may be as useful in the setting of acute brain injury as in chronic neurodegenerative disorders. An important route of further study will be to replicate these effects in human subjects. Such studies will help establish the safety and efficacy of intrathecal GBCAs in human subjects and provide valuable insights into their potential use in clinical settings.

b. Intravenous gadolinium enhancement of glymphatic T1 MRI:

The clinical usefulness of intrathecal gadolinium is limited by its invasiveness and the inherent risks of intrathecal injections. An alternative method, T1-weighted MRI with GBCAs administered intravenously, has been a topic of interest in recent years. Animal and human studies have thus far been targeted at understanding the path that GBCAs take from the blood to CSF in glymphatic circulation. In a study of six rats, intravenous injection of gadodiamide was found to coincide with an instantaneous increase in MRI signal intensity in the fourth ventricle, supporting the notion that GBCAs may immediately transfer from the blood to the CSF to reach the brain and glymphatic system [24]. A rapid change in T1 MRI values after the intravenous injection of a GBCA was also seen in a study of 25 healthy human subjects [25]. Taoka and Naganawa noted that GBCAs remain in glymphatic circulation significantly longer than they do in cerebral blood vessels, which was seen as a rapid decrease in the concentration of a GBCA due to renal elimination [26,27].

Another study using repetitive T1 MRI after GBCA injection revealed a sequential enhancement pattern beginning with blood vessels and peripheral tissues at 30 min post-injection and peaking in the grey matter, white matter, and ventricles at 90 min [28]. However, it remains to be seen whether T1 MRI with intravenous gadolinium is capable of detecting and characterizing glymphatic disruptions with the same efficacy as GBCAs via the intrathecal route. Much of the current animal studies on the glymphatic system use intrathecal administration of gadolinium over intravenous, which is possibly due to the greater ease of access to the cisterna magna in murine models. Consequently, there is currently a pressing need for more studies on intravenous gadolinium enhancement of T1 MRI in direct comparison with intrathecal gadolinium methods.

An intravenous GBCA was used to quantify leakage across the blood–brain barrier (BBB) in a study of healthy individuals age-matched to patients with early Alzheimer's disease. Leakage was higher in patients with Alzheimer's disease and correlated with significantly lower scores on the Mini-Mental State Examination [28]. In this way, glymphatic MRI enhanced with intravenous GBCAs may be used to estimate the severity of early Alzheimer's disease and may be another promising method to screen and prognosticate for

neurodegenerative diseases (Figure 2). Ultimately, more research is needed to fully understand the capabilities and limitations of gadolinium-enhanced MRI for glymphatic imaging.

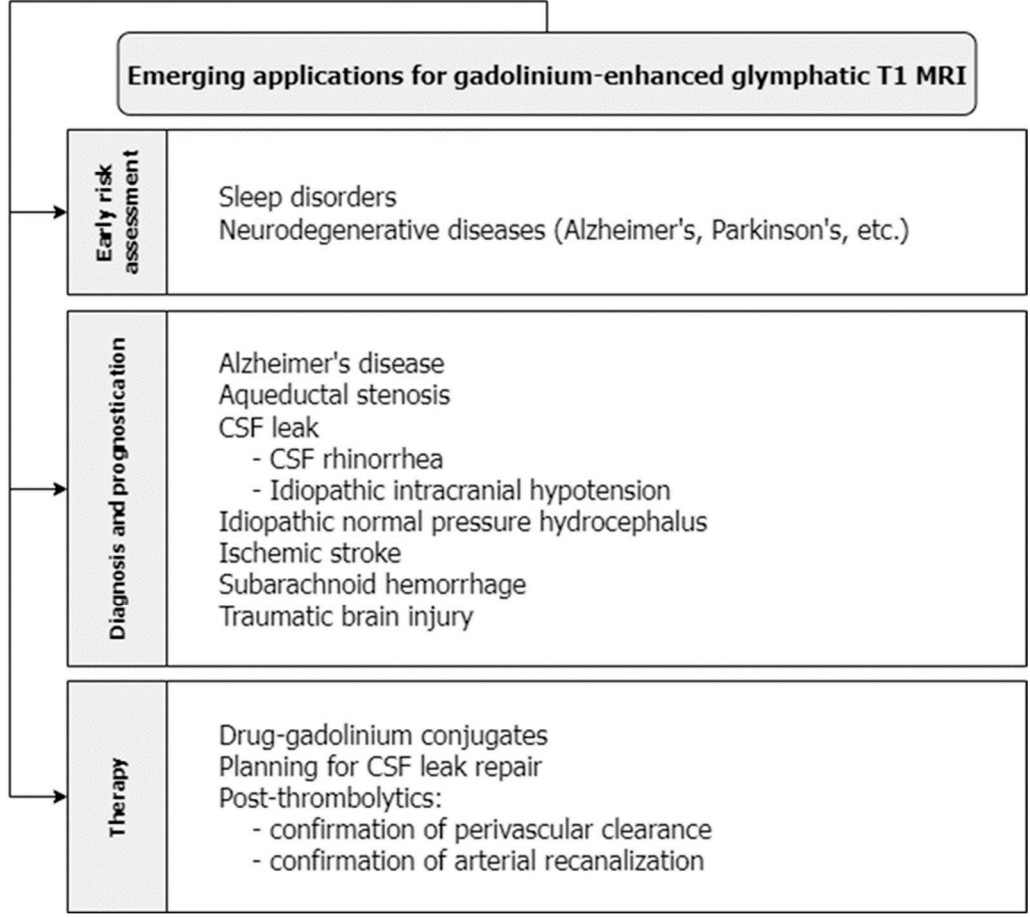

**Figure 2.** T1-weighted MRI with gadolinium contrast has emerging applications in screening for glymphatic disruption associated with sleep and neurodegenerative disorders, diagnosing acute and chronic neurological disorders, planning for the repair of CSF leakage, and evaluating the efficacy of therapeutic agents that act within the glymphatic system.

*2.2. Diffusion Tensor Imaging/Diffusion MR Technique/4D Flow MRI*

The dysregulation of the glymphatic system is also implicated in various neuroinflammatory and traumatic disease processes, including subdural hemorrhages and associated herniation syndromes [5,29]. Through diffusion MRI techniques, motion probing gradients (MPGs) that generate magnetic fields in various orientations can be used to assess the diffusion of glymphatic flow [5]. When these diffusion indices, denoted by the b value, are summated in 3D space, they are then visualized through diffusion tensor imaging (DTI) [5]. However, it should be noted that this technique solely maps the perivascular spaces (PVSs) and does not include extracranial drainage through the meninges or other routes. The concept of DTI is shown in the figure below (Figure 3); however, further research needs to be conducted to incorporate the entire glymphatic system into DTI mapping.

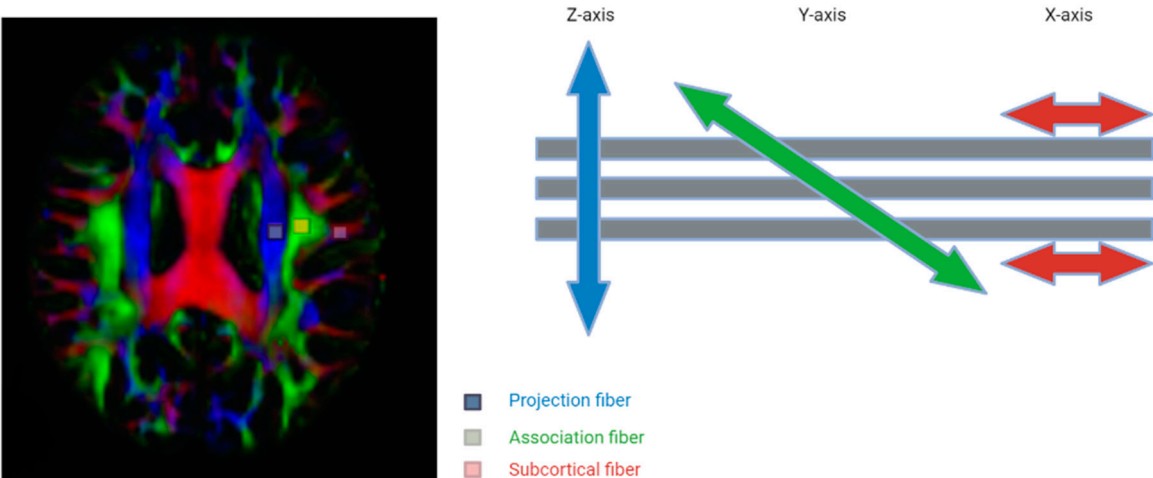

**Figure 3.** The concept of the diffusion tensor image analysis along the perivascular space (DTI-ALPS) method with 3 distinct regions of interest Adapted with permission from Taoka et al. *Jpn J Radiol* 2017, 35, 172–178, with permission of Japan Radiological Society [30].

There are several advantages to DTI over more traditional isotope tracer imaging modalities: While tracer studies have the limitation of relying on the tracer to spread temporally to distribute throughout the brain, DTI uses the magnetic spin of water molecules when MPGs are applied, providing a picture of the brain in real-time [5,31]. Similarly, DTI studies are also far less invasive, since tracer studies required tracer infusions administered intravenously or intrathecally.

Increased production of amyloid-β (Aβ) plaques and hyperphosphorylated tau tangles promote a pro-inflammatory state for the pathogenesis of Alzheimer's disease (AD). Dysfunction of the glymphatic system seems to contribute as it pertains to the clearance of neurotoxins from the interstitial fluid from the blood–brain barrier [31]. An impaired balance of increased production coupled with decreased clearance of the interstitial fluid (ISF) is well established in the pathogenesis of AD [32]. Additional studies have demonstrated that DTI is able to discern white matter lesions in the brains of patients with AD with greater sensitivity than standard MRI techniques [33,34]. Early pathologic changes seen in various types of dementia share a common inflammatory process of reactive astrogliosis and demyelination [34]. These changes can be evaluated using DTI as they reflect changes in the diffusion of water in these pathologies. A cohort analysis assessing a surrogate marker of cognition impairment in dementia found that median DTI diffusion values correlated with various levels of dementia [35].

Furthermore, impaired glymphatic clearance may also represent a potential diagnostic and therapeutic target for various neurotraumatic states. Some studies have shown that a portion of ISF clearance is a direct result of the bulk flow from the glymphatic system through aquaporin-4 (AQP4) channels transported from astrocytes [36,37]. The dysregulation of fluid shift following neurotrauma is the mechanism resulting in brain swelling and increased intracranial pressure (ICP). One study found decreased expression of AQP4 channels in the perivascular space as their localization in perivascular astrocyte foot processes following traumatic brain injury [38]. Thus, mapping the glymphatic system in such neurotraumatic instances can be very helpful.

During the acute stage of brain ischemia, there is a markedly decreased diffusion of water that is likely secondary to cytotoxic edema due to the dysregulation of the Na-K membrane pump and subsequent changes in the extracellular volume fraction [39]. Several studies have demonstrated how cellular edema resulting in the compromise of the blood–brain barrier can be safely visualized with diffusion-weighted MR imaging [22,40]. A study by Gaberel et al. demonstrated that glymphatic system function is severely impaired following a subarachnoid hemorrhage and following an ischemic stroke. In the same

study, it was also determined that the administration of a tissue plasminogen activator led to improvements in glymphatic flow [22]. Similarly, the cellular and vasogenic edema seen in patients at risk of acute stroke is difficult to visualize with conventional MRI. Diffusion-weighted imaging, however, distinguishes cellular and vasogenic edema and may prove therapeutic in discerning hyperacute ischemic encephalopathy associated with hypertension versus infarction, which has vastly different management and prognosis [41].

In addition to DTI, four-dimensional flow-sensitive MRI (4D flow MRI) refers to phase-contrast MRI using blood flow in all three directions represented in a spatial and temporal fashion and can provide blood flow characteristics, including its direction and velocity [42]. In a study of 50 patients diagnosed with ischemic strokes, a higher acute lesion volume measured in DTI directly correlated with worsened clinical severity measured with the NIH Stroke Scale [43]. The ability of DTI to detect hyperacute ischemia in minutes means that DTI may be used in conjunction with perfusion studies to extrapolate clinical severity and prognosis in stroke management.

As previously discussed, the assessment of intracranial 4D flow MRI allows for the concurrent evaluation of quantitative hemodynamics and vascular architecture. Intracranial hemodynamics is important in the pathogenesis of strokes, transient ischemic attacks, and vasospasms [44]. The addition of 4D flow with contrast-enhanced MR angiography allows for a greater signal-to-noise ratio and allows for the visualization of small vessels [45]. This makes 4D flow studies promising for clinicians in assessing the risk of stroke in patients with cardiovascular disease with greater sensitivity [45,46].

DTI is also useful for stroke management via the evaluation of CSF fluid in the perivascular space following a subarachnoid hemorrhage. This allows a clinician to predict the morbidity and damage following a stroke [30]. As discussed above, while both DTI and 4D flow MRI studies have great value in understanding the phenomenon of a dysfunctional glymphatic system following a hemorrhage, together, they are promising neuroimaging modalities that will be able to optimize the prognosticating and management of neuro-trauma and inflammatory disorders in various clinical settings.

### 2.3. SPECT/CT Imaging

According to a recent study conducted in mice, it has been discovered that subarachnoid hemorrhages have a disruptive effect on the inflow of cerebrospinal fluid (CSF) after a 24 h period [22]. Fibrin and fibrinogen deposits occlude the perivascular spaces and result in the inhibition of the glymphatic system [47]. In traumatic brain injuries, mouse studies have found that there is a decrease in glymphatic influx along with diminished clearance of radiotracers, persisting for over 28 days following the injury [4].

Single-photon emission computed tomography/computed tomography (SPECT/CT) has historically been used to quantify brain perfusion while assessing receptor binding in numerous diseases, such as dementia, vascular diseases, cancers, and neuropsychiatric diseases [48,49]. SPECT imaging detects gamma rays and is used clinically and preclinically [50]. Combining SPECT with CT allows for the detection of tracer distribution to be visualized in three dimensions with a higher specificity and sensitivity than MRI [51].

One study investigated the use of SPECT imaging compared with MRI to map the glymphatic system in rat models [51]. Additionally, the study compared ex vivo fluorescence with MRI and SPECT imaging [51]. All three imaging techniques demonstrated high amounts of tracer accumulation within the ventral brain regions; however, only ex vivo fluorescence detected tracer accumulation in the lateral cortex and indusium griseum [51]. Over time, the study evaluated the distribution of the tracer and found that MRI and SPECT imaging revealed similar patterns [51]. The tracer spread primarily to the ventral surface of the brain and then spread into the brain parenchyma as time progressed [51]. The study revealed that areas with high tracer distribution had an influx phase of 60 min, followed by a gradual clearance phase [51]. In areas with lower tracer concentrations, it took 120 min for the tracer to accumulate, followed by a gradual decrease [51].

The flow of CSF through the glymphatic system depends on the extent of CSF entry in the brain [52]. Thus, expansion of the interstitial space or increased access to the ISF is associated with an increase in glymphatic activity. Certain conditions like using anesthesia or sleeping trigger this response and thus increase CSF [53]. The study analyzed the effect of MRI and SPECT imaging to measure tracer amounts during these anesthesia events in drugs and found that there was a difference in the time–activity curves for the two imaging modalities when assessing high tracer concentrations [51]. Overall, the study in rats found that SPECT/CT imaging is a great option for studying the spatial and temporal aspects of the glymphatic system [51]. The researchers were able to use SPECT to determine the tracer flow velocity in the ventral compartment and were also able to analyze the extent of CSF shunting to the lymph nodes [51]. SPECT also demonstrated higher tracer specificity and quantitative signals compared with MRI within the animal study [51]. Moreover, because SPECT uses tracer concentrations, certain disadvantages like working with the deeper brain can be introduced [51]. Thus, tracer concentrations should always be modified based on specific goals.

Animal studies have demonstrated that SPECT/CT imaging may be useful in the study of neurodegenerative diseases linked to glymphatic system dysfunction. Currently, SPECT is used to determine the extent of amyloid-β accumulation in Alzheimer's [54,55]. Further research is needed to better understand CSF dynamics in humans, but SPECT imaging offers a promising route to study the glymphatic system and has demonstrated advantages over MRI.

Another study in rats found that the administration of hypertonic saline enhanced the glymphatic delivery of intrathecally administered agents [7]. Researchers found that hypertonic saline improved the brain-wide availability of administered small gold nanoparticles and also found that these particles were quickly removed from the body, decreasing the concern of renal toxicity [56]. The researchers used SPECT/CT imaging to quantify the extent of clearance of the tracer from the injection site while also determining the tracer distribution within the animal's body [7]. Compared with MRI, SPECT/CT is more sensitive and allows for smaller tracer volumes to be detected [7]. The study found that SPECT/CT imaging is a useful tool for visualizing and maximizing intrathecal CNS drug delivery [7]. However, there is very limited research on the use of SPECT/CT imaging in neurodegenerative diseases in humans with glymphatic system dysfunction and further research is needed to elucidate its role. However, it does offer a promising imaging modality for analyzing CSF flow and nanoparticle uptake in the glymphatic system.

### 2.4. Ultrafast MR Encephalography

Ultrafast magnetic resonance encephalography (MREG) can produce an image of the brain within 100 milliseconds [8]. Combining ultrafast MREG with cardiorespiratory monitoring allows for whole-brain coverage and allows for the analysis of all physiological pulsations [57]. Human studies using ultrafast MREG have found that there are three distinct pulsation mechanisms that maintain the flow of CSF and elimination of waste products within the brain: very low frequency (<0.1 Hz), respiratory (0.2–0.3 Hz), and cardiac (0.8–1.2 Hz) (Figure 4) [58]. The primary contributor to bulk CSF flow and exchange of CSF with interstitial fluid is believed to be arterial pulsations that occur during cardiac systole, allowing for solutes to move from the peri-arterial spaces into the extracellular brain tissue [59,60].

The cardiovascular pulsation creates a negative MREG signal impulse and then generates a positive charge within the cortex [8]. This pulsation begins at the basal peri-arterial spaces near the circle of Willis and then travels to the cortex [8]. Within the cortex, the respiratory pulsation is prominent and also extends into the peri-venous collection system [8]. The last pulsation is via the slow vasomotor waves seen via very low-frequency and low-frequency waves [8]. These slow vasomotor waves create unique spatiotemporal patterns [8].

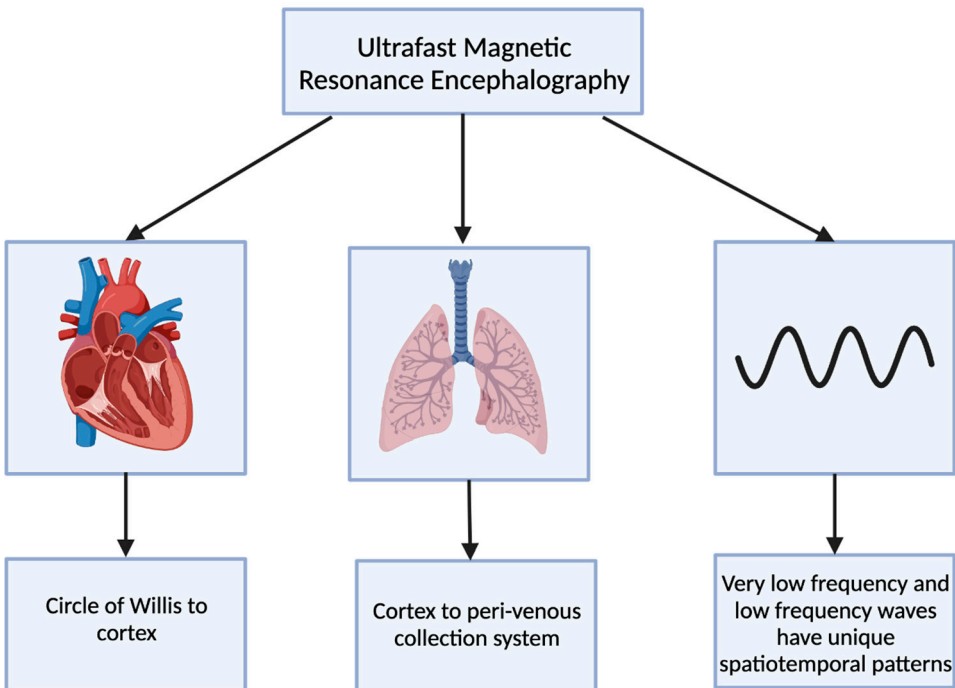

**Figure 4.** The 3 distinct pulsations of the glymphatic system captured using ultrafast magnetic resonance encephalography.

In neurodegenerative diseases, impaired glymphatic clearance and drainage are believed to be dysfunctional; therefore, there is poor removal of protein debris [8]. Ultrafast MREG is capable of capturing glymphatic pulsation mechanisms and may be useful in elucidating the mechanisms resulting in glymphatic impairment and protein buildup in neurodegenerative conditions [36]. Ultrafast MREG may be useful in understanding the mechanism of glymphatic system dysfunction in neurodegenerative diseases as it can capture the distinct pulsations produced by the glymphatic system. However, further research is needed in which ultrafast MREG is used to map the glymphatic system in human and animal studies.

*2.5. Fluorescence Microscopy*

Fluorescence microscopy is a technique used to monitor cell physiology by making use of fluorescent agents to help increase the image contrast and spatial resolution of a given specimen [61]. These fluorescent molecules, known as fluorophores or fluorochromes, undergo photoluminescence, where portions of light are absorbed and emitted by the fluorophore. These signals of light are detected with a camera system that allows for enhanced imaging under the microscope [62].

The glymphatic system has been associated with diseases which are caused by the accumulation of certain compounds. Utilizing fluorophores to attach to pathologic solutes of interest can allow the visualization of these solutes and their presence in and clearance from the glymphatic system [63]. Utilization of this technique was demonstrated in one study regarding Alzheimer's disease in mice. The visualization of amyloid plaques in the brains of mice demonstrated a reduction in glymphatic influx and clearance of amyloid plaques as the mice aged, displaying how the concentration and clearance of proteins within the brain can be observed [4].

Fluorophores were also used in mice to map the glymphatic system by injecting the mice with fluorescently labeled dextrans. The mice dextrans were injected into the cerebrospinal fluid and were visualized traveling into the perivascular spaces of surface arteries and then moving deeper into penetrating arteries. Furthermore, in Tie2-GFP:NG2-DsRed double reporter mice, the fluorescent tracer i.e., the ovalbumin passed through the

astrocyte foot-processes and glial-limiting membranes by entering through the perivascular space between the smooth muscle cells. Following an injection of fluorescent tracer, the arteries, veins, capillaries were able to be directly identified [64]. The use of fluorescent tracers validated the perivascular influx of CSF and the directionality of this fluid movement with CSF entering the brain exclusively via periarterial spaces and ISF leaving the brain via perivenous channels [4,64].

Two-photon fluorescence laser-scanning microscopy, a variation of fluorescence microscopy, is a technique that can facilitate live high-resolution spatiotemporal imaging. Two-photon fluorescence laser-scanning microscopy can be used to obtain the visualization of the whole-cortex vascular 3D structure, as well as to understand the blood flow in the capillary network of the brain. It is also useful in measuring the size, speed, and flux of structures within the brain and the cranial window. This imaging technique allows for a deeper understanding of space and time regarding the images taken, and thus providing direct evidence regarding the pathophysiology uncovering after a neurotrauma such as an ischemic stroke [63,65].

Fluorescent microscopy provides powerful advanced imaging, allows for real-time evaluation of the presence and flow of desired molecules through the brain, and allows for multidimensional analysis of brain structure and pathophysiological mechanisms of neurotrauma and disease [65–69].

### 2.6. Macroscopic Imaging Techniques

Macroscopic imaging techniques include the utilization of macroscopic CSF tracers, transcranial macroscopic imaging, and fluorescence macroscopy. These imaging techniques utilize the use of fluorophores to bind to compounds of interest and provide quantitative information on compound concentrations and distribution through optically turbid media, such as tissue (Figure 5). The use of this technology allows for the possibility to study and analyze intact organ structures and provides information about how therapies can be more optimally administered to reach their target destination, and it can be used for guidance during operations such as brain tumor resections [69,70].

Transcranial macroscopic imaging has been used to noninvasively evaluate the in vivo delivery pathways of CSF fluorescent tracers that were shown to be distributed through the glymphatic system. Transcranial macroscopic imaging was able to determine that CSF tracer entry increased by three-fold without disruption of the BBB when the plasma osmolality was increased. Additionally in a mouse model, it showed that increasing the plasma osmolality reversed the inhibited glymphatic flow and suppression, which is a characteristic of the awake state of Alzheimer's disease. Lastly, it showed that an increased plasma osmolality also showed a five-fold increase in the binding of antibodies to amyloid-β (Aβ) plaques, via enhancing its delivery. The utilization of a combination of macroscopic CSF tracers and transcranial macroscopic imaging spearheaded the possibility to visualize and study how to improve the penetration of therapeutic antibodies to the CNS [10,71–74].

Real-time image-guided surgical tumor excisions promise to improve clinical outcomes and prolong the lives of patients. Histopathological analysis of rat brains ex vivo showed the location and invasion of the tumor within the deep structures of the brain [70,75]. Characteristics of tumors such as cell type and degree of vascularization can also be observed and studied [70,75]. Advanced-stage glioma is the most aggressive form of malignant brain tumor with a short survival time [76]. A study evaluating time-resolved laser-induced fluorescence spectroscopy was able to classify low-grade gliomas with 100% sensitivity and 98% specificity. This provides a potentially valuable tool for neurosurgeon neuropathology teams to rapidly distinguish between tumor and normal brain tissue during surgery [77].

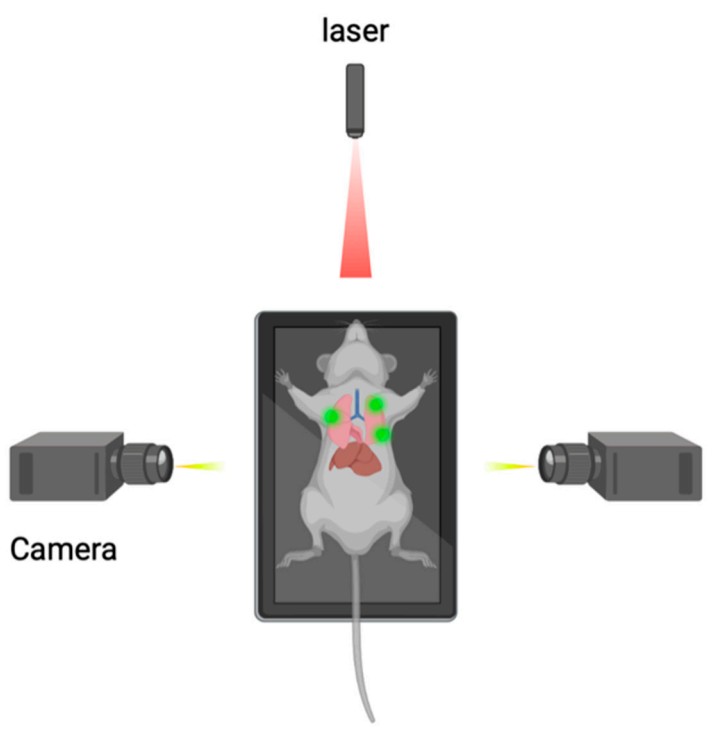

**Figure 5.** Detection of compounds of interest through the utilization of fluorophores.

### 3. Conclusions

The glymphatic system is a recently discovered waste clearance system that has been associated with many diseases including Alzheimer's disease, hemorrhage, and neurotrauma. Thus, it opens an array of research opportunities to improve and understand their prognoses. Currently, ex vivo fluorescence microscopy of brain slices, MRI, and macroscopic cortical imaging are the most common ways of determining glymphatic system function. However, as discussed above, to improve future translational efforts, the development of a minimally invasive and safe imaging approach is necessary. Further research should also focus on comparing the depth, speed, and other parameters of these imaging methods in the setting of glymphatic system mapping.

**Author Contributions:** Conceptualization, B.L.-W.; Writing—original draft preparation, R.N.T., I.P.K., I.G., D.T.F., P.S. and E.M.V.; Writing—reviewing & editing, R.N.T., I.P.K., I.G., D.T.F., P.S. and E.M.V. All authors have read and agreed to the published version of the manuscript.

**Funding:** This research received no external funding.

**Institutional Review Board Statement:** Not applicable.

**Data Availability Statement:** No new data were created or analyzed in this study. Data sharing is not applicable to this article.

**Conflicts of Interest:** The authors declare no conflict of interest. Pratiksha Sharma is employed by Thermo Fisher Scientific Inc. The paper reflects the views of the scientist, and not the company.

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
