# Peer review of "Mapping the Glymphatic Pathway Using Imaging Advances"

_2571-8800, doi:10.3390/j6030031_

Round 1

Reviewer 1 Report

Review: ‘Mapping the Glymphatic Pathway Using Imaging Advances”

Summary: In this review article, the author endeavors to examine the significant progress made in preclinical imaging techniques for elucidating the glymphatic system pathways.

This topic holds significant value as the glymphatic system is a relatively novel field that has the potential to uncover new drug targets. Given the importance of gaining a deeper understanding of this system, this review aims to investigate various imaging modalities used to study it.

However, it is crucial to address the identified major flaws in the review, and the manuscript must be rewritten. The major flaws in the manuscript are  mentioned below:

Comments:

The initial paragraph's content (lines 22-32) offers an overly simplistic explanation of the glymphatic system, which might lead readers to confusion regarding its functioning. The authors should place emphasis on the significance of Perivascular spaces (PVS) and the involvement of AQP4 water channels in facilitating the influx of cerebrospinal fluid (CSF) into the interstitial fluid (ISF). Furthermore, it is important for the authors to elaborate on the disposal pathways for metabolically loaded ISF, including the role of meningeal lymphatics as extracranial exit routes.

In the second paragraph (35-39), the authors assert that gadolinium-based contrast agents (GBCAs) can readily pass through the blood-brain barrier (BBB), whereas it remains unclear which classical MRI agents the authors are specifically referring to. It should be noted that GBCAs have been utilized in clinical imaging for several decades.

Line 47: “The glymphatic flow is a key feature of the glymphatic system”, Statement is confusing. Key features are PVS, AQP4, paravenous exit routes, etc

Line 104-106: Authors quotes Illif’s article regarding convective and diffusive flow in different brain compartments. Authors should also mention the Molecular Weight independent convective flow in PVS and SAS regions.

Line 193-222: The authors discuss the applications of the DTI-ALPS technique in glymphatic mapping; however, they do not present additional evidence of its use in the interstitial fluid (ISF). It is important to note that this technique solely maps the perivascular spaces (PVS) and does not encompass the entire glymphatic system or the extracranial drainage through the meninges or other routes.

Line 224-235: After discussing the DTI-ALPS technique, the authors deviate to a topic that is unrelated and fails to establish a clear connection with the preceding paragraphs. It is important for the authors to provide a coherent explanation of how this new topic is connected to the previously discussed concepts.

Figure 3. “ The concept of the diffusion tensor image analysis along with the perivascular 276 space (DTI-ALPS) method with 3 distinct regions of interest” Source of this figure should be mentioned.

Author Response

The initial paragraph's content (lines 22-32) offers an overly simplistic explanation of the glymphatic system, which might lead readers to confusion regarding its functioning. The authors should place emphasis on the significance of Perivascular spaces (PVS) and the involvement of AQP4 water channels in facilitating the influx of cerebrospinal fluid (CSF) into the interstitial fluid (ISF). Furthermore, it is important for the authors to elaborate on the disposal pathways for metabolically loaded ISF, including the role of meningeal lymphatics as extracranial exit routes.

Additional details have been added regarding the AQP4 channels which helps emphasize the significance of the PVS. Details of drainage via lymphatic vessels, including meningeal lymphatic vessels, have also been added to highlight the drainage route.

In the second paragraph (35-39), the authors assert that gadolinium-based contrast agents (GBCAs) can readily pass through the blood-brain barrier (BBB), whereas it remains unclear which classical MRI agents the authors are specifically referring to. It should be noted that GBCAs have been utilized in clinical imaging for several decades.

This statement has been removed due to incorrect conclusions made from the reading. A clarifying statement instead has been added that GBCA is important in mapping CSF.

Line 47: “The glymphatic flow is a key feature of the glymphatic system”, Statement is confusing. Key features are PVS, AQP4, paravenous exit routes, etc

The glymphatic flow is facilitated via the PVS, AQP4, etc features of the glymphatic system. Glymphatic flow is important however it is

Line 104-106: Authors quotes Illif’s article regarding convective and diffusive flow in different brain compartments. Authors should also mention the Molecular Weight independent convective flow in PVS and SAS regions.

A statement was added to acknowledge the finding that convective flow of gadolinium-based contrast agents in the proximal perivascular channels and subarachnoid space occurs independently of molecular weight.

Line 193-222: The authors discuss the applications of the DTI-ALPS technique in glymphatic mapping; however, they do not present additional evidence of its use in the interstitial fluid (ISF). It is important to note that this technique solely maps the perivascular spaces (PVS) and does not encompass the entire glymphatic system or the extracranial drainage through the meninges or other routes.

This clarification has been made and stated in the paper

Line 224-235: After discussing the DTI-ALPS technique, the authors deviate to a topic that is unrelated and fails to establish a clear connection with the preceding paragraphs. It is important for the authors to provide a coherent explanation of how this new topic is connected to the previously discussed concepts.

We rearranged this paragraph so it can flow better. Previously we discussed how mapping the glymphatic system can be helpful in AD, and now we discuss why it can be helpful in other neurodegenerative states. 

Figure 3. “ The concept of the diffusion tensor image analysis along with the perivascular 276 space (DTI-ALPS) method with 3 distinct regions of interest” Source of this figure should be mentioned.

Thank you for pointing this out. The source has been cited.

Reviewer 2 Report

In this review article, the authors discuss the current ways to map the glymphatic system, and address the advances being made from preclinical mapping. I support the publication of this manuscript with minor revisions after the following concerns are addressed.

1. The article introduces several imaging methods of glial lymphatic pathways in detail, but there are no resulting images, so that readers cannot intuitively understand the imaging methods through the images;

2. The article analyzes the advantages of several imaging methods, but does not focus on comparing the depth, speed and other parameters of several imaging methods;

3. It is suggested that the authors cite recent advances in photoacoustic imaging in the glial lymphatic system in the background

[1] Shaykevich S, Chan R W, Rana C, et al. Optoacoustic imaging of the glymphatic system[J]. Veins and Lymphatics, 2022, 11(1).

[2] Wang Z, Yang F, Shi W, et al. Monitoring the perivascular cerebrospinal fluid dynamics of the glymphatic pathway using co-localized photoacoustic microscopy[J]. Optics Letters, 2023, 48(9): 2265-2268.

Minor editing of English language required

Author Response

The article introduces several imaging methods of glial lymphatic pathways in detail, but there are no resulting images so readers cannot intuitively understand the imaging methods through the images

For this suggestion, would siting the images that each imaging methods produce be helpful (ex: figure 3)? 

The article analyzes the advantages of several imaging methods but does not focus on comparing the depth, speed, and other parameters of several imaging methods

I researched but currently, no studies compare the parameters of different imaging methods for mapping the glymphatic system in particular. Moreover, I have added this point to the conclusion as a future research option.

3. It is suggested that the authors cite recent advances in photoacoustic imaging in the glial lymphatic system in the background

Done

[1] Shaykevich S, Chan R W, Rana C, et al. Optoacoustic imaging of the glymphatic system[J]. Veins and Lymphatics, 2022, 11(1).

[2] Wang Z, Yang F, Shi W, et al. Monitoring the perivascular cerebrospinal fluid dynamics of the glymphatic pathway using co-localized photoacoustic microscopy[J]. Optics Letters, 2023, 48(9): 2265-2268.

Round 2

Reviewer 1 Report

Comments and Suggestions are as follows:

Lines 27-36: These perivascular systems express aquaporin-4 water channels (AQP4) to facilitate the transportation of nutrients. With the help of AQP4 water channels, the exchange of cerebrospinal fluid (CSF) and into interstitial fluid (ISF) through aquaporins.occurs to assist in lymphatic drainage and waste removal. This exchange allows  proper elimination and reabsorption of solutes and metabolites, fluid and electrolyte balance, and regulation of internal body pressure. The drainage is driven in the direction of  CSF into the sinuses of the brain’s dura mater. Lymphatic vessels, including meningeal  lymphatic vessels, in the dura mater absorbs CSF and ISF through the glymphatic system  and transport the fluid to the base of the skull for extracranial drainage.

Please add the significance of astrocytic end feet, an example is as follows:

The most important component of this glymphatic system includes the perivascular spaces ( PVS) which surround the penetrating arteries. These perivascular spaces are surrounded by astrocytic end feet which has aqp4 water channels assisting the entry of CSF into the brain ISF from PVS. The CSF-ISF mix full of metabolic waste drains out of CNS via perivenous space or along cranial and spinal nerves. The waste ultimately is disposed of via meningeal lymphatics to the extracranial lymphatic network.

The above statement can be supported by any review article by Dr. Nedergaard’s work.

Figure 3. The concept of the diffusion tensor image analysis along with the perivascular  space (DTI-ALPS) method with 3 distinct regions of interest [47].

I could not find the image in the ref number 47, but found similar image in ref number 46.

Please cross check the references

Line 298: In current mouse studies, subarachnoid hemorrhages disrupt CSF inflow after 24 hours

Suggested edits: According to a recent study conducted on mice, it has been discovered that subarachnoid hemorrhages have a disruptive effect on the inflow of cerebrospinal fluid (CSF) after a 24-hour period.

Line 326- There is increased CSF during sleep or when patients are given ketamine/xylazine or ketamine/dexmedetomidine as anesthesia [53,54].

Rewrite this statement as this is confusing: Instead author may emphasize on how these anesthesia helps CSF gaining access to ISF, thereby activating glymphatic system.

Line 345- 357: Author explains the advantages of CT/SPECT over MRI. Please add its disadvantages for the well-balanced review.

minor editing is required.

Author Response

1. Lines 27-36: These perivascular systems express aquaporin-4 water channels (AQP4) to facilitate the transportation of nutrients. With the help of AQP4 water channels, the exchange of cerebrospinal fluid (CSF) and into interstitial fluid (ISF) through aquaporins.occurs to assist in lymphatic drainage and waste removal. This exchange allows  proper elimination and reabsorption of solutes and metabolites, fluid and electrolyte balance, and regulation of internal body pressure. The drainage is driven in the direction of  CSF into the sinuses of the brain’s dura mater. Lymphatic vessels, including meningeal  lymphatic vessels, in the dura mater absorbs CSF and ISF through the glymphatic system  and transport the fluid to the base of the skull for extracranial drainage.

Please add the significance of astrocytic end feet

I've added the significance as suggested. Thank you for suggesting this and hope it flows better.

2. Figure 3. The concept of the diffusion tensor image analysis along with the perivascular  space (DTI-ALPS) method with 3 distinct regions of interest [47].

I could not find the image in the ref number 47, but found similar image in ref number 46.

Please cross check the references

Sorry about this inconvenience. This reference has been changed and the others have been checked.

3. Line 298: In current mouse studies, subarachnoid hemorrhages disrupt CSF inflow after 24 hours

Suggested edits: According to a recent study conducted on mice, it has been discovered that subarachnoid hemorrhages have a disruptive effect on the inflow of cerebrospinal fluid (CSF) after a 24-hour period.

This suggestion has been applied

4. Line 326- There is increased CSF during sleep or when patients are given ketamine/xylazine or ketamine/dexmedetomidine as anesthesia [53,54].

Rewrite this statement as this is confusing: Instead author may emphasize on how these anesthesia helps CSF gaining access to ISF, thereby activating glymphatic system.

I've changed this sentence and also added the suggested information

5. Line 345- 357: Author explains the advantages of CT/SPECT over MRI. Please add its disadvantages for the well-balanced review.

This makes sense. The disadvantage of using tracer concentration (in SPECT) has been added.
